# RRL: Resnet as representation for Reinforcement Learning

Rutav Shah[1] and Vikash Kumar[2,3]

*Abstract*— **Generalist robots capable of performing dexterous, contact-rich manipulation tasks will enhance productivity and provide care in un-instrumented settings like homes. Such tasks warrant operations in real-world only using the robot's proprioceptive sensor such as onboard cameras, joint encoders, etc which can be challenging for policy learning owing to the high dimensionality and partial observability issues. We propose RRL: Resnet as representation for Reinforcement Learning – a straightforward yet effective approach that can learn complex behaviors directly from proprioceptive inputs. RRL fuses features extracted from pre-trained Resnet into the standard reinforcement learning pipeline and delivers results comparable to learning directly from the state. In a simulated dexterous manipulation benchmark, where the state of the art methods fails to make significant progress, RRL delivers contact rich behaviors. The appeal of RRL lies in its simplicity in bringing together progress from the fields of Representation Learning, Imitation Learning, and Reinforcement Learning. Its effectiveness in learning behaviors directly from visual inputs with performance and sample efficiency matching learning directly from the state, even in complex high dimensional domains, is far from obvious.**

## I. INTRODUCTION

Recently, Reinforcement learning (RL) has seen tremendous momentum and progress [9, 19, 37, 21] in learning complex behaviors from states [18, 24, 17]. Most success stories, however, are limited to simulations or instrumented laboratory conditions as real world doesn't provide direct access to its internal state. Not only learning with state-space, but visual observation spaces have also found reasonable success [26, 42]. However, the majority of these methods have been tested on low-dimensional, 2D tasks [31] that lack depth information. Contact rich manipulation tasks, on the other hand, are high dimensional and necessitate intricate details in order to be completed successfully. In order to deliver the promise presented by data-driven techniques, we need efficient techniques that can learn complex behaviors unobtrusively without the need for environment instrumentation.

Learning without environment instrumentation, especially in unstructured settings like homes, can be quite challenging [59, 34, 46]. Challenges include – (a) Decision making with incomplete information owing to partial observability as the agents must rely only on proprioceptive on-board sensors (vision, touch, joint position encoders, etc) to perceive and act. (b) The influx of sensory information makes the input space quite high dimensional. (c) Information contamination due to sensory noise and task-irrelevant conditions like lightning,

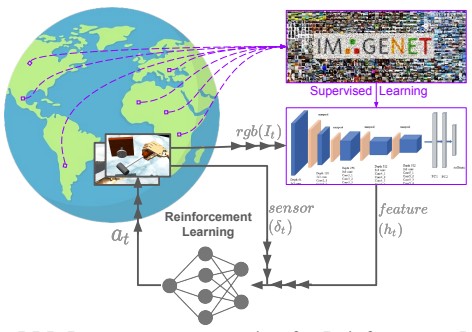

Fig. 1. RRL Resnet as representation for Reinforcement Learning takes a small step in bridging the gap between Representation learning and Reinforcement learning. RRL pre-trains an encoder on a wide variety of real world classes like ImageNet dataset using a simple supervised classification objective. Since the encoder is exposed to a much wider distribution of images while pretraining, it remains effective whatever distribution the policy might induce during the training of the agent. This allows us to freeze the encoder after pretraining without any additional efforts.

shadows, etc. (d) Most importantly, the scene being flushed with information irrelevant to the task (background, clutter, etc). Agents learning under these constraints is forced to take a large number of samples simply to untangle these task-irrelevant details before it makes any progress on the true task objective. A common approach to handle these high dimensionality and multi-modality issues is to learn representations that distil information into low dimensional features and use them as inputs to the policy. While such ideas have found reasonable success [43, 40], designing such representations in a supervised manner requires a deep understanding of the problem and domain expertise. An alternative approach is to leverage unsupervised representation learning to autonomously acquire representations based on either reconstruction [13, 59, 56] or contrastive [51, 52] objective. These methods are quite brittle as the representations are acquired from narrow task-specific distributions [61], and hence, do not generalize well across different tasks Table I. Additionally, they acquire task-specific representations, often needing additional samples from the environment leading to poor sample efficiency or domains specific data-augmentations for training representations.

The key idea behind our method stems from an intuitive observation over the desiderata of a good representation i.e. (a) it should be low dimensional for a compact representation. (b) it should be able to capture silent features encapsulating the diversity and the variability present in a real-world task for better generalization performance. (c) it should be robust to irrelevant information like noise, lighting, viewpoints, etc so that it is resilient to the changes in surroundings. (d) it should provide effective representation in the entire distribution that a policy can induce for effective learning. These requirements

[1]Department of Computer Science and Engineering, Indian Institute of Technology, Kharagpur, India `rutavms@gmail.com`
[2]Department of Computer Science, University of Washington, Seattle, USA `vikash@cs.washington.edu`
[3]Facebook AI Research, USA

are quite harsh needing extreme domain expertise to manually design and an abundance of samples to automatically acquire. Can we acquire this representation without any additional effort? Our work takes a very small step in this direction.

The key insight behind our method (Figure 1) is embarrassingly simple – representations do not necessarily have to be trained on the exact task distribution; a representation trained on a *sufficiently* wide distribution of real-world scenarios, will remain effective on any distribution a policy optimizing a task in the real world might induce. While training over such wide distribution is demanding, this is precisely what the success of large image classification models [8, 10, 54, 12] in Computer Vision delivers – representations learned over a large family of real-world scenarios.

**Our Contributions**: We list the major contributions

1) We present a surprisingly simple method (RRL) at the intersection of representation learning, imitation learning (IL) and reinforcement learning (RL) that uses features from pre-trained image classification models (Resnet34) as representations in standard RL pipeline. Our method is quite general and can be incorporated with minimal changes to most state based RL/IL algorithms.

2) Task-specific representations learned by supervised as well as unsupervised methods are usually brittle and suffer from distribution mismatch. We demonstrate that features learned by image classification models are general towards different task (Figure 2), robust to visual distractors, and when used in conjunction with standard IL and RL pipelines can efficiently acquire policies directly from proprioceptive inputs.

3) While competing methods have restricted results primarily to planar tasks devoid of depth perspectives, on a rich collection of simulated high dimensional dexterous manipulation tasks, where state-of-the-art methods struggle, we demonstrate that RRL can learn rich behaviors directly from visual inputs with performance & sample efficiency approaching state-based methods.

4) Additionally, we underline the performance gap between the SOTA approaches and RRL on simple low dimensional tasks as well as high dimensional more realistic tasks. Furthermore, we experimentally establish that the commonly used environments for studying image based continuous control methods are not a true representative of real-world scenario.

## II. RELATED WORK

RRL rests on recent developments from the fields of Representation Learning, Imitation Learning and Reinforcement Learning. In this section, we outline related works leveraging representation learning for visual reinforcement and imitation learning.

### A. Learning without explicit representation

A common approach is to learn behaviors in an end to end fashion – from pixels to actions – without explicit distinction between feature representation and policy

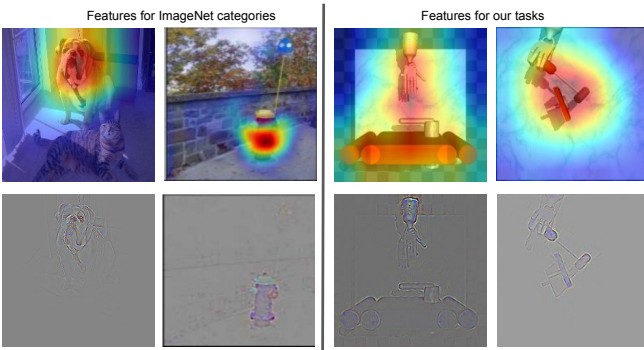

Fig. 2. Visualization of Layer 4 of Resnet model of the top 1 class using Grad-CAM [45][Top] and Guided Backpropogation [11][Bottom]. This indicates that Resnet is indeed looking for the right features in our task images (right) in spite of such high distributional shift.

representations. Success stories in this categories range from seminal work [5] mastering Atari 2600 computer games using only raw pixels as input, to [14] which learns trajectory-centric local policies using Guided Policy Search [4] for diverse continuous control manipulation tasks in the real world learned directly from camera inputs. More recently, [35] has demonstrated success in acquiring multi-finger dexterous manipulation [33] and agile locomotion behaviors using off-policy action critic methods [24]. While learning directly from pixels has found reasonable success, it requires training large networks with high input dimensionality. Agents require a prohibitively large number of samples to untangle task-relevant information in order to acquire behaviors, limiting their application to simulations or constrained lab settings. RRL maintains an explicit representation network to extract low dimensional features. Decoupling representation learning from policy learning delivers results with large gains in efficiency. Next, we outline related works that use explicit representations.

### B. Learning with supervised representations

Another approach is to first acquire representations using expert supervision, and use features extracted from representation as inputs in standard policy learning pipelines. A predominant idea is to learn representative keypoints encapsulating task details from the input images and using the extracted keypoints as a replacement of the state information [38]. Using these techniques, [43, 39] demonstrated tool manipulation behaviors in rich scenes flushed with task-irrelevant details. [41] demonstrated simultaneous manipulation of multiple objects in the task of Baoding ball tasks on a high dimensional dexterous manipulation hand. Along with the inbuilt proprioceptive sensing at each joint, they use an RGB stereo image pair that is fed into a separate pre-trained tracker to produce 3D position estimates [57] for the two Baoding balls. These methods, while powerful, learn task-specific features and requires expert supervision, making it harder to (a) translate to variation in tasks/environments, and (b) scale with increasing task diversity. RRL, on the other hand, uses single task-agnostic representations with better generalization capability making it easy to scale.

## C. Learning with unsupervised representations

With the ambition of being scalable, this group of methods intends to acquire representation via unsupervised techniques. [30] uses contrastive learning to time-align visual features across different embodiment to demonstrate behavior transfer from human to a Fetch robot. [20], [62, 59] use variational inference [7, 20] to learn compressed latent representations and use it as input to standard RL pipeline to demonstrate rich manipulation behaviors. [47] additionally learns dynamics models directly in the latent space and use model-based RL to acquire behaviors on simulated tasks. On similar tasks, [36] uses multi-step variational inference to learn world dynamic as well as rewards models for off-policy RL. [51] use image augmentation with variational inference to construct features to be used in standard RL pipeline and demonstrate performance at par with learning directly from the state. [49, 48] demonstrate comparable results by assimilating updates over features acquired only via image augmentation. Similar to supervised methods, unsupervised methods often learns task-specific brittle representations as they break when subjected to small variations in the surroundings and often suffers challenges from non-stationarity arising from the mismatch between the distribution representations are learned on and the distribution policy induces. To induce stability, RRL uses pre-trained stationary representations trained on distribution with wider support than what policy can induce. Additionally, representations learned over a wide distribution of real-world samples are robust to noise and irrelevant information like lighting, illumination, etc.

## D. Learning with representations and demonstrations

Learning from demonstrations has a rich history. We focus our discussion on DAPG [17], a state-based method which optimizes for the natural gradient [2] of a joint loss with imitation as well as reinforcement objective. DAPG has been demonstrated to outperform competing methods [15, 16] on the high dimensional ADROIT dexterous manipulation task suite we test on. RRL extends DAPG to solve the task suite directly from proprioceptive signals with performance and sample efficiency comparable to state-DAPG. Unlike DAPG which is on-policy, FERM [58] is a closely related off-policy actor-critic methods combining learning from demonstrations with RL. FERM builds on RAD [49] and inherits its challenges like learning task-specific representations. We demonstrate via experiments that RRL is more stable, more robust to various distractors, and convincingly outperforms FERM since RRL uses a fixed feature extractor pre-trained over wide variety of real world images and avoids learning task specific representations.

## III. BACKGROUND

RRL solves a standard Markov decision process (Section III-A) by combining three fundamental building blocks - (a) Policy gradient algorithm (Section III-B), (b) Demonstration bootstrapping (Section III-C), and (c) Representation learning (Section III-D). We briefly outline these fundamentals before detailing our method in Section IV.

## A. Preliminaries: MDP

We model the control problem as a Markov decision process (MDP), which is defined using the tuple: $\mathcal{M} = (\mathcal{S}, \mathcal{A}, \mathcal{R}, \mathcal{T}, \rho_0, \gamma)$. $\mathcal{S} \in \mathbb{R}^n$ and $\mathcal{A} \in \mathbb{R}^m$ represent the state and actions. $\mathcal{R} : \mathcal{S} \times \mathcal{A} \to \mathbb{R}$ is the reward function. In the ideal case, this function is simply an indicator for task completion (*sparse reward setting*). $\mathcal{T} : \mathcal{S} \times \mathcal{A} \to \mathcal{S}$ is the transition dynamics, which can be stochastic. In model-free RL, we do not assume any knowledge about the transition function and require only sampling access to this function. $\rho_0$ is the probability distribution over initial states and $\gamma \in [0, 1)$ is the discount factor. We wish to solve for a stochastic policy of the form $\pi : \mathcal{S} \times \mathcal{A} \to \mathbb{R}$ which optimizes the expected sum of rewards:

$$\eta(\pi) = \mathbb{E}_{\pi, \mathcal{M}} \left[ \sum_{t=0}^{\infty} \gamma^t r_t \right] \qquad (1)$$

## B. Policy Gradient

The goal of the RL agent is to maximise the expected discounted return $\eta(\pi)$ (Equation 1) under the distribution induced by the current policy $\pi$. Policy Gradient algorithms optimize the policy $\pi_\theta(a \mid s)$ directly, where $\theta$ is the function parameter by estimating $\nabla \eta(\pi)$. First we introduce a few standard notations, Value function : $V^\pi(s)$, Q function : $Q^\pi(s, a)$ and the advantage function : $A^\pi(s, a)$. The advantage function can be considered as another version of Q-value with lower variance by taking the state-value off as the baseline.

$$V^\pi(s) = \mathbb{E}_{\pi \mathcal{M}} \left[ \sum_{t=0}^{\infty} \gamma^t r_t \mid s_0 = s \right]$$
$$Q^\pi(s, a) = \mathbb{E}_{\mathcal{M}} \left[ \mathcal{R}(s, a) \right] + \mathbb{E}_{s' \sim \mathcal{T}(\int, \dashv)} \left[ V^\pi(s') \right]$$
$$A^\pi(s, a) = Q^\pi(s, a) - V^\pi(s)$$
$$(2)$$

The gradient can be estimated using the Likelihood ratio approach and Markov property of the problem [1] and using a sampling based strategy,

$$\nabla \eta(\pi) = g = \frac{1}{NT} \sum_{i=0}^{N} \sum_{t=0}^{T} \nabla_\theta \ log \ \pi_\theta(a_t^i | s_t^i) \hat{A}^\pi(s_t^i, a_t^i, t)$$
$$(3)$$

Amongst the wide collection of policy gradient algorithms, we build upon Natural Policy Gradient (NPG) [2] to solve our MDP formulation owing to its stability and effectiveness in solving complex problems. We refer to [32] for a detailed background on different policy gradient approaches. In the next section, we describe how human demonstrations can be effectively used along with NPG to aid policy optimization.

## C. Demo Augmented Policy Gradient

Policy Gradients with appropriately shaped rewards can solve arbitrarily complex tasks. However, real-world environments seldom provide shaped rewards, and it must be manually specified by domain experts. Learning with sparse signals, such as task completion indicator functions, can relax

domain expertise in reward shaping but it results in extremely high sample complexity due to exploration challenges. DAPG ([17]) combines policy gradients with few demonstrations in two ways to mitigate this issue and effectively learn from them. We represent the demonstration dataset using $\rho_D = \left\{ \left( s_t^{(i)}, a_t^{(i)}, s_{t+1}^{(i)}, r_t^{(i)} \right) \right\}$ where $t$ indexes time and $i$ indexes different trajectories.

(1) Warm up the policy using few demonstrations (25 in our setting) using a simple Mean Squared Error(MSE) loss, i.e, initialize the policy using behavior cloning [Eq 4]. This provides an informed policy initialization that helps in resolving the early exploration issue as it now pays attention to task relevant state-action pairs and thereby, reduces the sample complexity.

$$L_{BC}(\theta) = \frac{1}{2} \sum_{i,t \in minibatch} \left( \pi_\theta(s_t^{(i)}) - a_t^{(i)H} \right)^2 \quad (4)$$

where, $\theta$ are the agent parameters and $a_t^{(i)H}$ represents the action taken by the human/expert.

(2) DAPG builds upon on-policy NPG algorithm [2] which uses a normalized gradient ascent procedure where the normalization is under the Fischer metric.

$$\theta_{k+1} = \theta_k + \sqrt{\frac{\delta}{g^T \hat{F}_{\theta_k}^{-1} g}} \hat{F}_{\theta_k}^{-1} g \quad (5)$$

where $\hat{F}_{\theta_k}$ is the Fischer Information Metric at the current iterate $\theta_k$,

$$\hat{F}_{\theta_k} = \frac{1}{T} \sum_{t=0}^{T} \nabla_\theta log \ \pi_\theta(a_t|s_t) \nabla_\theta log \ \pi_\theta(a_t|s_t)^T \quad (6)$$

and $g$ is the sample based estimate of the policy gradient [Eq 3]. To make the best use of available demonstrations, DAPG proposes a joint loss $g_{aug}$ combining task as well as imitation objective. The imitation objective asymptotically decays over time allowing the agent to learn behaviors surpassing the expert.

$$g_{aug} = \sum_{(s,a) \in \rho_\pi} \nabla_\theta \ ln \ \pi_\theta(a|s) A^\pi(s, a)$$
$$+ \sum_{(s,a) \in \rho_D} \nabla_\theta \ ln \ \pi_\theta(a|s) w(s, a) \quad (7)$$

where, $\rho_\pi$ is the dataset obtained by executing the current policy, $\rho_D$ is the demonstration data and $w(s, a)$ is the heuristic weighting function defined as :

$$w(s, a) = \lambda_0 \lambda_1^k \max_{(s', a') \in \rho_\pi} A^\pi(s', a') \quad \forall \ (s, a) \in \rho_D \quad (8)$$

DAPG has proven to be successful in learning policy for the dexterous manipulation tasks with reasonable sample complexity.

### D. Representation Learning

DAPG has thus far only been demonstrated to be effective with access to low-level state information which is not readily available in real-world. DAPG is based on NPG which works well but faces issues with input dimensionality and hence, cannot be directly used with the input images acquired from onboard cameras. Representation learning [6] is learning representations of input data typically by transforming it or extracting features from it, which makes it easier to perform the task (in our case it can be used in place of the exact state of the environment). Let $I \in \mathbb{R}^n$ represents the high dimensional input image, then

$$h = f_\rho(I) \quad (9)$$

where $f$ represents the feature extractor, $\rho$ is the distribution over which $f$ is valid and $h \in \mathbb{R}^d$ with $d << n$ is the compact, low dimensional representation of $I$. In the next section, we outline our method that scales DAPG to solve directly from visual information.

### IV. RRL: RESNET AS REPRESENTATION FOR RL

In an ideal RL setting, the agent interacts with the environment based on the current state, and in return, the environment outputs the next state and the reward obtained. This works well in a simulated environment but in a real-world scenario, we do not have access to this low-level state information. Instead we get the information from cameras ($I_t$) and other onboard sensors like joint encoders ($\delta_t$). To overcome the challenges associated with learning from high dimensional inputs, we use representations that project information into a lower-dimensional manifolds. These representations can be (a) learned in tandem with the RL objective. However, this leads to non-stationarity issue where the distribution induced by the current policy $\pi_i$ may lie outside the expressive power of $f$, $\pi_i \not\subset \rho_i$ at any step $i$ during training. (b) decoupled from RL by pre-training $f$. For this to work effectively, the feature extractor must be trained on a sufficiently wide distribution such that it covers any distribution that the policy might induce during training, $\pi_i \subset \rho \ \forall \ i$. Getting hold of such task specific training data beforehand becomes increasingly difficult as the complexity and diversity of the task increases. To this end, we propose to use a fixed feature extractor (Section V-B) that is pretrained on a wide variety of real world scenarios like ImageNet dataset [Highlighted in *purple* in Figure 1]. We experimentally demonstrate that the diversity (Section V-C) of the such feature extractor allows us to use it across all tasks we considered. The use of pre-trained representations induces stability to RRL as our representations are frozen and do-not face the non-stationarity issues encountered while learning policy and representation in tandem.

The features ($h_t$) obtained from the above feature extractor are appended with the information obtained from the internal joint encoders of the Adroit Hand ($\delta_t$). As a substitute of the exact state ($s_t$), we empirically show that $[h_t, \delta_t]$ can be used as an input to the policy. In principle any RL algorithm can be deployed to learn the policy, in RRL we build upon Natural Policy Gradients [3] owing to effectiveness in solving

**Algorithm 1** RRL

1: **Input:** 25 Human Demonstrations $\rho_D$
2: **Initialize** using Behavior Cloning [Eq.4].
3: **repeat**
4:     **for** i = 1 to n **do**
5:         **for** t = 1 to horizon **do**
6:             Take action
7:     $a_t = \pi_\theta([Encoder(I_t), \delta_t])$
8:     and receive $I_{t+1}, \delta_{t+1}, r_{t+1}$
9:     from the environment.
10:         **end for**
11:     **end for**
12:     Compute $\nabla_\theta \, log \, \pi_\theta(a_t|s_t)$ for each $(s, a) \in \rho_\pi, \rho_D$
13:     Compute $A^\pi(s, a)$ for each $(s, a) \in \rho_\pi$ and $w(s, a)$
      for each $(s, a) \in \rho_D$ according to Equations 2, 8
14:     Calculate policy gradient according to 7
15:     Compute Fisher matrix 6
16:     Take the gradient ascent step according to 5.
17:     Update the parameters of the value function in order
      to approximate(2) : $V_k^\pi(s_t^{(n)}) \approx \sum_{t'=t}^{T} \gamma^{t'-t} r_t^{(n)}$
18: **until** Satisfactory performance

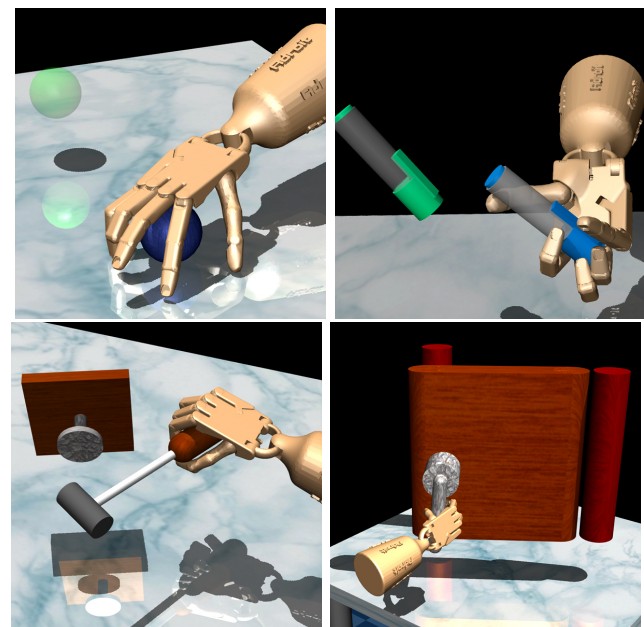

Fig. 3. ADROIT manipulation suite consisting of complex dexterous manipulation tasks involving object relocation, in hand manipulation (pen repositioning), tool use (hammering a nail), and interacting with human centric environments (opening a door).

complex high dimensional tasks [17]. We present our full algorithm in Algorithm-1.

## V. EXPERIMENTAL EVALUATIONS

Our experimental evaluations aims to address the following questions: (1) Does pre-tained representations acquired via large real world image dataset allow RRL to learn complex tasks directly from proprioceptive signals (camera inputs and joint encoders)? (2) How does RRL's performance and efficiency compare against other state-of-the-art methods? (3) How various representational choices influence the generality and versatility of the resulting behaviors? (5) What are the effects of various design decisions on RRL? (6) Are commonly used benchmarks for studying image based continuous control methods effective?

### A. Tasks

Applicability of prior proprioception based RL methods [49, 48, 47] have been limited to simple low dimensional tasks like Cartpole, Cheetah, Reacher, Finger spin, Walker, Ball in cup, etc. Moving beyond these simple domains, we investigate RRL on Adroit manipulation suite [17] which consists of contact-rich high-dimensional dexterous manipulation tasks (Figure 3) that have found to be challenging ever for state ($s_t$) based methods. Furthermore, unlike prior task sets, which are fundamentally planar and devoid of depth perspective, the Adroit manipulation suite consists of visually-rich physically-realistic tasks that demand representations untangling complex depth information.

### B. Implementation Details

We use standard Resnet-34 model as RRL's feature extractor. The model is pre-trained on the ImageNet dataset which consists of 1000 classes. It is trained on 1.28 million

images on the classification task of ImageNet. The last layer of the model is removed to recover a 512 dimensional feature space and all the parameters are frozen throughout the training of the RL agent. During inference, the observations obtained from the environment are of size 256 × 256, a center crop of size 224 × 224 is fed into the model. We also evaluate our model using different Resnet sizes (Figure 7). All the hyperparameters used for training are summarized in Appendix(Table II). We report an average performance over three random seeds for all the experiments.

### C. Results

In Figure 4, we contrast the performance of RRL against the state of the art baselines. We begin by observing that NPG [3] struggles to solve the suite even with full state information, which establishes the difficulty of our task suite. DAPG(State) [17] uses privileged state information and a few demonstrations from the environment to solve the tasks and pose as the best case oracle. RRL demonstrates good performance on all the tasks, relocate being the hardest, and often approaches performance comparable to our strongest oracle-DAPG(State).

A competing baseline FERM [1] [58] is quite unstable in these tasks. It starts strong for hammer and door tasks but saturates in performance. It makes slow progress in pen, and completely fails for relocate. In Figure 5 [Left] we compare the computational footprint of FERM (along with other methods, discussed in later sections) with RRL. We note that our method not only outperforms FERM but also is approximately five times more compute-efficient.

---
[1]Reporting best performance amongst over 30 configurations per task we tried in consultation with the FERM authors.

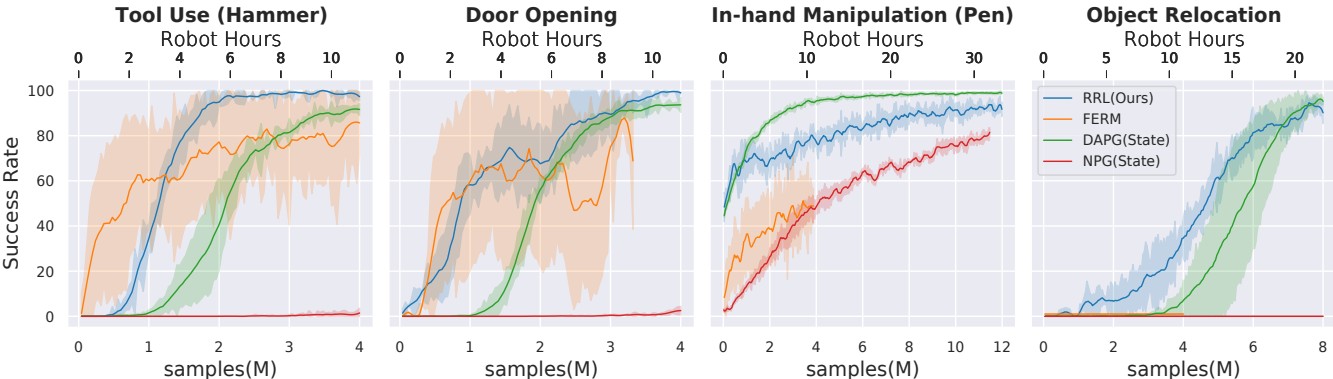

Fig. 4. Performance on ADROIT dexterous manipulation suite [17]: State of the art policy gradient method NPG(State) [29] struggles to solve the suite even with privileged low level state information, establishing the difficulty of the suite. Amongst demonstration accelerated methods, RRL(Ours) demonstrates stable performance and approaches performance of DAPG(State) [17] (upper bound), a demonstration accelerated method using privileged state information. A competing baseline FERM [58] makes good initial, but unstable, progress in a few tasks and often saturates in performance before exhausting our computational budget (40 hours/ task/ seed).

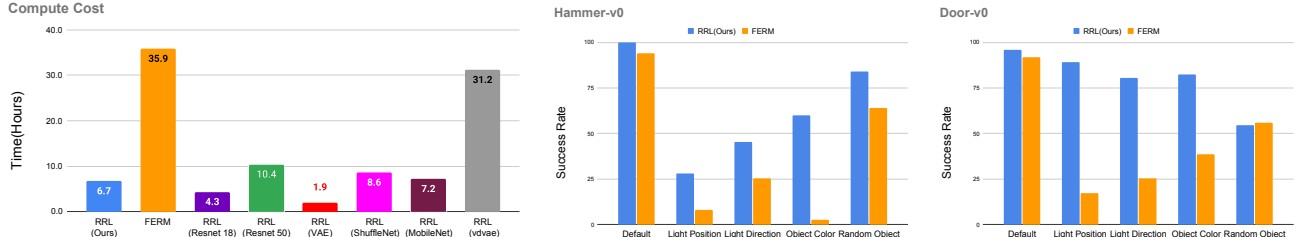

Fig. 5. LEFT: Comparison of the computational cost of RRL with Resnet34 i.e RRL(Ours), FERM - Strongest baseline, RRL with Resnet 18, RRL with Resnet 50, RRL(VAE), RRL with ShuffleNet, RRL with MobileNet and RRL with Very Deep VAE baseline. CENTER,RIGHT: Influence of various environment distractions (lightning condition, object color) on RRL(Ours), and FERM. RRL(Ours) consistently performs better than FERM in all the variations we considered.

### D. Effects of Visual Distractors

In Figure 5 [Center, Right] we probe the robustness of the final policies by injecting visual distractors in the environment during inference. We note that the resilience of the resnet features induces robustness to RRL's policies. On the other hand, task-specific features learned by FERM are brittle leading to larger degradation in performance. In addition to improved sample and time complexity resulting from the use of pre-trained features, the resilience, robustness, and versatility of Resnet features lead to policies that are also robust to visual distractors, clutter in the scene. More details about the experiment setting is provided in Section VII-H in Appendix.

### E. Effect of Representation

**Is Resnet lucky?** To investigate if architectural choice of Resnet is lucky, in Figure 6 we test different models pretrained on ImageNet dataset as RRL's feature extractors – MobileNetV2 [44], ShuffleNet [27] and state of the art hierarchical VAE [60] [Refer Section VII-E in Appendix for more details]. Not much degradation in performance is observed with respect to the Resnet model. This highlights that it is not the architecture choices in particular, rather the dataset on which models are being pre-trained, that delivers generic features effective for the RL agents.

**Task-specific vs Task-agnostic representation:** In Figure 7, we compare the performance between (a) learning task

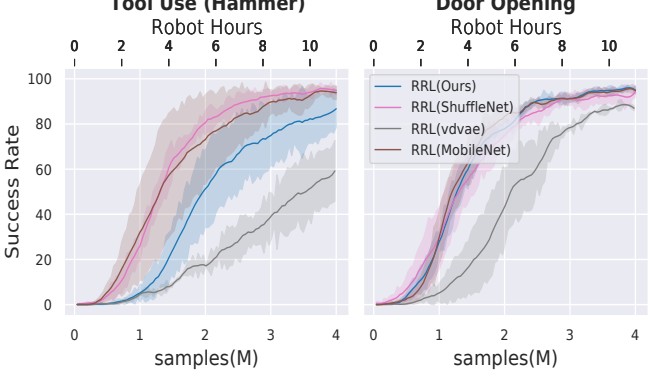

Fig. 6. Effect of different types of Feature extractor pretrained on ImageNet dataset, highlighting that not just Resnet but any feature extractor pretrained on a sufficiently wide distribution of data remains effective.

specific representations (VAE) (b) generic representation trained on a very wide distribution (Resnet). We note that RRL using Resnet34 significantly outperforms a variant RRL(VAE) (see appendix for details Section VII-G) that learns features via commonly used variational inference techniques on a task specific dataset [22, 23, 25, 28]. This indicates that pre-trained Resnet provides task agnostic and superior features compared to methods that explicitly learn *brittle* (Section-V-H) and task-specific features using additional samples from the environment. It is important to note that the latent dimension of the Resnet34 and VAE are kept same (512) for a fair comparison, however, the

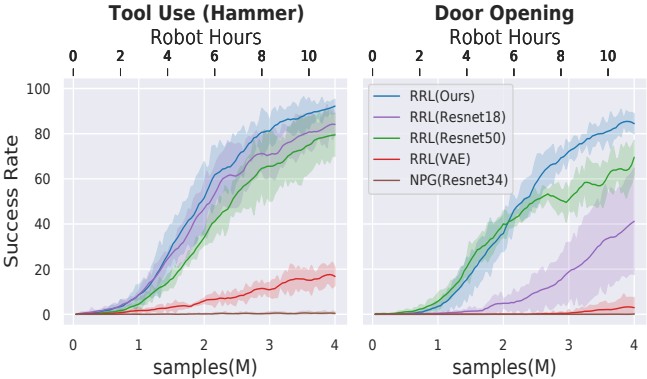

Fig. 7. Influence of representation: RRL(Ours), using resnet34 features, outperforms commonly used representation (RRL(VAE)) learning method VAE. Amongst different Resnet variations, Resnet34 strikes the balance between representation capacity and computational overhead. NPG(Resnet34) showcases the performance with Resnet34 features but without demonstration bootstrapping, indicating that only representational choices are not enough to solve the task suite.

model sizes are different as one operates on a very wide distribution while the other on a much narrower task specific dataset. Additionally, we summarize the compute cost of both the methods RRL(Ours) and RRL(VAE) in Figrue 5 [Left]. We notice that even though RRL(VAE) is the cheapest, its performance is quite low (Figure 7). RRL(Ours) strikes a balance between compute and efficiency.

### F. Effects of proprioception choices and sensor noise

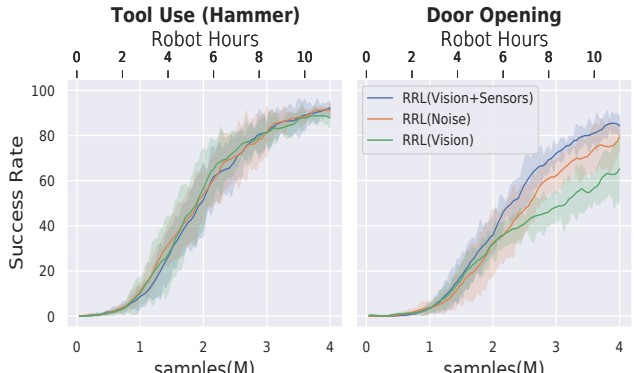

Fig. 8. Influence of proprioceptive signals on RRL(Vision+sensors-Ours): RRL(Noise) demonstrates that RRL remains effectiveness in presence of noisy (2%) proprioception. RRL(Vision) demonstrates that RRL remains performant with (only) visual inputs as well.

While it's hard to envision a robot without proprioceptive joint sensing, harsh conditions of the real-world can lead to noisy sensing, even sensor failures. In Figure 8, we subjected RRL to (a) signals with 2% noise in the information received from the joint encoders RRL(Noise), and (b) only visual inputs are used as proprioceptive signals RRL(Vision). In both these cases, our methods remained performant with slight to no degradation in performance.

### G. Ablations and Analysis of Design Decisions

In our next set of experiments, we evaluate the effect of various design decisions on our method. In Figure 7, we study the effect of different Resnet features as our representation. Resnet34, though computationally more demanding (Figure 5) than Resnet18, delivers better performance owing to its

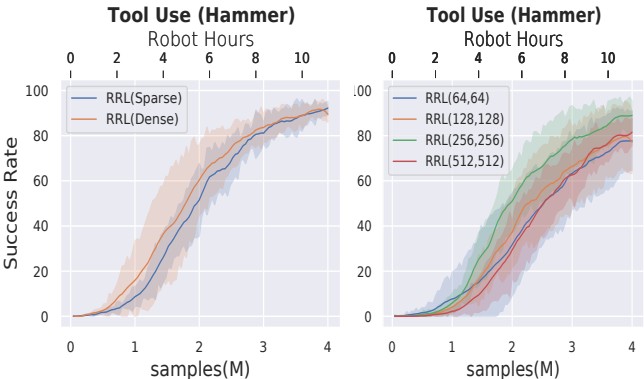

Fig. 9. LEFT: Influence of rewards signals: RRL(Ours), using sparse rewards, remains performant with a variation $RRL_{dense}$ using well-shaped dense rewards. RIGHT: Effect of policy size on the performance of RRL . We observe that it is quite stable with respect to a wide range of policy sizes.

improved representational capacity and feature expressivity. A further boost in capacity (Resnet50) degrades performance, likely due to the incorporation of less useful features and an increase in samples required to train the resulting larger policy network.

Reward design, especially for complex high dimensional tasks, requires domain expertise. RRL replaces the needs of well-shaped rewards by using a few demonstrations (to curb the exploration challenges in high dimensional space) and sparse rewards (indicating task completion). This significantly lowers the domain expertise required for our methods. In Figure 9-LEFT, we observe that RRL (using sparse rewards) delivers competitive performance to a variant of our methods that uses well-shaped dense rewards while being resilient to variation in policy network capacity (Figure 9-RIGHT).

### H. Rethinking benchmarking for visual RL

DMControl [31] is a widely used benchmark for proprioception based RL methods – RAD [49], SAC+AE [56], CURL [51], DrQ [48]. While these methods perform well (Table I) on such simple DMControl tasks, their progress struggles to scale when met with task representative of real world complexities such as realistic Adroit Manipulation benchmarks (Figure 4).

For example we demonstrate in Figure 4 that a representative SOTA methods FERM (uses expert demos along with RAD) struggles to perform well on Adroit Manipulation benchmark. On the contrary, RRL using Resnet features pretrained on real world image dataset, delivers state comparable results on Adroit Manipulation benchmark while struggles on the DMControl (RRL+SAC: RRL using SAC and Resnet34 features I). This highlights large domain gap between the DMControl suite and the real-world.

We further note that the pretrained features learned by SOTA methods aren't as widely applicable. We use a pretrained RAD encoder (pretrained on Cartpole) as fixed feature extractor (Fixed RAD encoder in Table I) and retrain the policy using these features for all environments. The performance degrades for all the tasks except Cartpole. This

| 500K Step Scores | RRL+SAC | RAD | Fixed RAD Encoder | CURL | SAC+AE | State SAC |
|---|---|---|---|---|---|---|
| Finger, Spin | $422 \pm 102$ | $\mathbf{947} \pm 101$ | $789 \pm 190$ | $926 \pm 45$ | $884 \pm 128$ | $923 \pm 211$ |
| Cartpole, Swing | $357 \pm 85$ | $863 \pm 9$ | $\mathbf{875} \pm 01$ | $845 \pm 45$ | $735 \pm 63$ | $848 \pm 15$ |
| Reacher, Easy | $382 \pm 299$ | $\mathbf{955} \pm 71$ | $53 \pm 44$ | $929 \pm 44$ | $627 \pm 58$ | $923 \pm 24$ |
| Cheetah, Run | $154 \pm 23$ | $\mathbf{728} \pm 71$ | $203 \pm 31$ | $518 \pm 28$ | $550 \pm 34$ | $795 \pm 30$ |
| Walker, Walk | $148 \pm 12$ | $\mathbf{918} \pm 16$ | $182 \pm 40$ | $902 \pm 43$ | $847 \pm 48$ | $948 \pm 54$ |
| Cup, Catch | $447 \pm 132$ | $\mathbf{974} \pm 12$ | $719 \pm 70$ | $959 \pm 27$ | $794 \pm 58$ | $974 \pm 33$ |
| 100K Step Scores | | | | | | |
| Finger, Spin | $135 \pm 67$ | $\mathbf{856} \pm 73$ | $655 \pm 104$ | $767 \pm 56$ | $740 \pm 64$ | $811 \pm 46$ |
| Cartpole, Swing | $192 \pm 19$ | $828 \pm 27$ | $\mathbf{840} \pm 34$ | $582 \pm 146$ | $311 \pm 11$ | $835 \pm 22$ |
| Reacher, Easy | $322 \pm 285$ | $\mathbf{826} \pm 219$ | $162 \pm 40$ | $538 \pm 233$ | $274 \pm 14$ | $746 \pm 25$ |
| Cheetah, Run | $72 \pm 63$ | $\mathbf{447} \pm 88$ | $188 \pm 20$ | $299 \pm 48$ | $267 \pm 24$ | $616 \pm 18$ |
| Walker, Walk | $63 \pm 07$ | $\mathbf{504} \pm 191$ | $106 \pm 11$ | $403 \pm 24$ | $394 \pm 22$ | $891 \pm 82$ |
| Cup, Catch | $261 \pm 57$ | $\mathbf{840} \pm 179$ | $533 \pm 148$ | $769 \pm 43$ | $391 \pm 82$ | $746 \pm 91$ |

TABLE I

RESULTS ON DMCONTROL BENCHMARK. RAD OUTPERFORMS ALL THE BASELINES WHEREAS RRL PERFORMS WORSE IN THE 100K AND 500K ENVIRONMENTAL STEP BENCHMARK SUGGESTING THAT IT IS QUICKER TO LEARN TASK SPECIFIC REPRESENTATION IN SIMPLE TASKS WHEREAS FIXED RAD ENCODER HIGHLIGHTS THAT THE REPRESENTATIONS LEARNED BY RAD ARE NARROW AND TASK SPECIFIC.

highlights that the representation learned by RAD (even with various image augmentations) are task specific and fail to generalize to other tasks set with similar visuals. Furthermore, learning such task specific representations are easier on simpler scenes but their complexity grows drastically as the complexity of tasks and scenes increases. To ensure that important problems aren't overlooked, we emphasise the need for the community to move towards benchmarks representative of realistic real world tasks.

## VI. STRENGTHS, LIMITATIONS & OPPORTUNITIES

This paper presents an intuitive idea bringing together advancements from the fields of representation learning, imitation learning, and reinforcement learning. We present a very simple method named RRL that leverages Resnet features as representation to learn complex behaviors directly from proprioceptive signals. The resulting algorithm approaches the performance of state-based methods in complex ADROIT dexterous manipulation suite.

**Strengths**: The strength of our insight lies in its simplicity, and applicability to almost any reinforcement or imitation learning algorithm that intends to learn directly from high dimensional proprioceptive signals. We present RRL , an instantiation of this insight on top of imitation + (on-policy) reinforcement learning methods called DAPG, to showcase its strength. It presents yet another demonstration that features learned by Resnet are quite general and are broadly applicable. Resnet features trained over 1000s of real-world images are more robust and resilient in comparison to the features learned by methods that learn representation and policies in tandem using only samples from the task distribution. The use of such general but frozen representations in conjunction with RL pipelines additionally avoids the non-stationary issues faced by competing methods that simultaneously optimizes reinforcement and representation objectives, leading to more stable algorithms. Additionally, not having to train your own features extractors results in a significant sample and compute gains, Refer to Figure 5.

**Limitations**: While this work demonstrates promises of using pre-trained features, it doesn't investigate the data mismatch problem that might exist. Real-world datasets used to train resnet features are from human-centric environments. While we desire robots to operate in similar settings, there are still differences in their morphology and mode of operations. Additionally, resent (and similar models) acquire features from data primarily comprised of static scenes. In contrast, embodied agents desire rich features of dynamic and interactive movements.

**Opportunities**: RRL uses a single pre-trained representation for solving all the complex and very different tasks. Unlike the domains of vision and language, there is a non-trivial cost associated with data in robotics. The possibility of having a standard shared representational space opens up avenues for leveraging data from various sources, building hardware-accelerated devices using feature compression, low latency and low bandwidth information transmission.

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

# VII. APPENDIX

## A. Project's webpage

Full details of the project (including video results, codebase, etc) are available at https://sites.google.com/view/abstractions4rl.

## B. Overview of all methods used in baselines and ablations

The environmental setting and the feature extractor used in all the variations and different methods considered is summarized in Table VII-B

| | Observation | | | Latent Features | Demos | Rewards |
|---|---|---|---|---|---|---|
| | Vision (RGB) | Joint Encoders | Environment State | | | |
| RRL(Ours) | ✓ | ✓ | | Resnet34 | ✓ | Sparse |
| RRL(Resnet18) | ✓ | ✓ | | Resnet18 | ✓ | Sparse |
| RRL(Resnet50) | ✓ | ✓ | | Resnet50 | ✓ | Sparse |
| RRL (VAE) | ✓ | ✓ | | VAE | ✓ | Sparse |
| RRL(Vision) | ✓ | | | Resnet34 | ✓ | Sparse |
| FERM | ✓ | ✓ | | | ✓ | Sparse |
| NPG(State) | | ✓ | ✓ | | | Sparse |
| NPG(Vision) | ✓ | | | Resnet34 | | Sparse |
| DAPG(State) | | ✓ | ✓ | | ✓ | Sparse |
| RRL(Sparse) | ✓ | ✓ | | Resnet34 | ✓ | Sparse |
| RRL(Dense) | ✓ | ✓ | | Resnet34 | ✓ | Dense |
| RRL(Noise) | ✓ | ✓ | | Resnet34 | ✓ | Sparse |
| RRL(Vision + Sensors) | ✓ | ✓ | | Resnet34 | ✓ | Sparse |
| RRL(ShuffleNet) | ✓ | ✓ | | ShuffleNet-v2 | ✓ | Sparse |
| RRL(MobileNet) | ✓ | ✓ | | MobileNet-v2 | ✓ | Sparse |
| RRL(vdvae) | ✓ | ✓ | | Very Deep VAE | ✓ | Sparse |

## C. RRL(Ours)

| Parameters | Setting |
|---|---|
| BC batch size | 32 |
| BC epochs | 5 |
| BC learning rate | 0.001 |
| Policy Size | (256, 256) |
| vf _batch_size | 64 |
| vf_epochs | 2 |
| rl_step_size | 0.05 |
| rl_gamma | 0.995 |
| rl_gae | 0.97 |
| lam_0 | 0.01 |
| lam_1 | 0.95 |

TABLE II

HYPERPARAMETER DETAILS FOR ALL THE **RRL** VARIATIONS.

Same parameters are used across all the tasks (Pen, Door, Hammer, Relocate, PegInsertion, Reacher) unless explicitly mentioned. Sparse reward setting is used in all the hand manipulation environments as proposed by Rajeswaran et al. along with 25 expert demonstrations. We have directly used the parameters (summarize in Table II) provided by DAPG without any additional hyperparameter tuning except for the policy size (used same across all tasks). On the Adroit Manipulation task, 200 trajectories for Hammer-v0, Door-v0, Relocate-v0 whereas 400 trajectories for Pen-v0 per iteration are collected in each iteration.

## D. Results on MJRL Environment

We benchmark the performance of RRL on two of the MJRL environments [50], Reacher and Peg Insertion in Figure 10. These environments are quite low dimensional (7DoF Robotic arm) compared to the Adroit hand (24 DoF) but still require rich understanding of the task. In the peg insertion task, RRL delivers state comparable (DAPG(State)) results and significantly outperforms FERM. However, in the Reacher task, we notice that DAPG(State) and FERM perform surprisingly well whereas RRL struggles to perform initially. This highlights that using task specific representations in simple, low dimensional environments might be beneficial as it is easy to overfit the feature encoder for the task in hand while the Resnet features are quite generic. For the MJRL environment, shaped reward setting is used as provided in the repository [2] along with 200 expert demonstrations. For the Peg Insertion task 200 trajectories and for Reacher task 400 trajectories are collected per iteration.

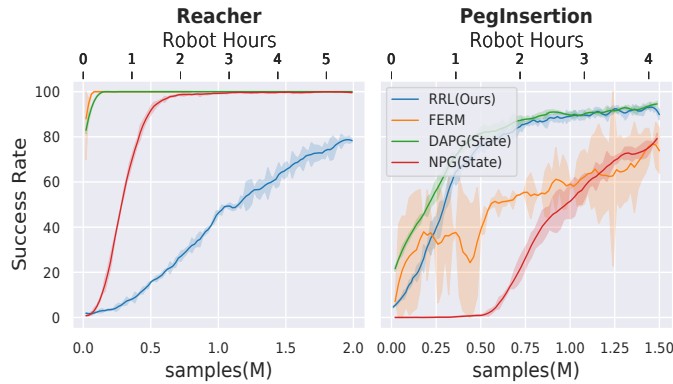

Fig. 10. **Results on MJRL Environment.** RRL outperforms FERM and delivers results on par with DAPG(State) in the PegInsertion task. In Reacher, FERM outperforms RRL following that learning task specific representations is easier in simple tasks.

### E. Other variations of RRL

a) **RRL(MobileNet), RRL(ShuffleNet)** : The encoders (ShuffleNet [27] and MobileNet [44]) are pretrained on ImageNet Dataset using a classification objective. We pick the pretrained models from torchvision directly and freeze the parameters during the entire training of the RL agent. Similar to RRL(Ours), the last layer of the model is removed and a latent feature of dimension 1024 and 1280 in case of ShuffleNet and MobileNet respectively is used.

b) **RRL(vdvae)** : We use a very recent state of the art hierarchical VAE [60] that is trained on ImageNet dataset. The code along with the pretrained weights are publically available [3] by the author. We use the intermediate features of the encoder of dimension 512. All the parameters are frozen similar to RRL(Ours).

### F. DMControl Experiment Details

For the RAD [49], CURL [51], SAC+AE [56] and State SAC [35], we report the numbers directly provided by Laskin et al. For SAC+RRL, Resnet34 is used as a fixed feature extractor and the past three output features (frame_stack= 3) are used as a representative of state information in SAC algorithm. For the fixed RAD encoder, we train the RL agent along with RAD encoder using the default hyperparameters provided by the authors for Cartpole environment. We used the trained encoder as a fixed feature extractor and retrain the policies for all the tasks. The frame_skip values are task specific as mentioned in [56] also outlined in Table IV. The hyperparameters used are summarized in the Table III where a grid search is made on actor_lr = $\{1e-3, 1e-4\}$, critic_lr = $\{1e-3, 1e-4\}$, critic_update_freq = $\{1, 2\}$, critic_tau = $\{0.01, 0.05, 0.1\}$ and an average over 3 seeds is reported. SAC implementation in PyTorch courtesy [55].

### G. RRL(VAE)

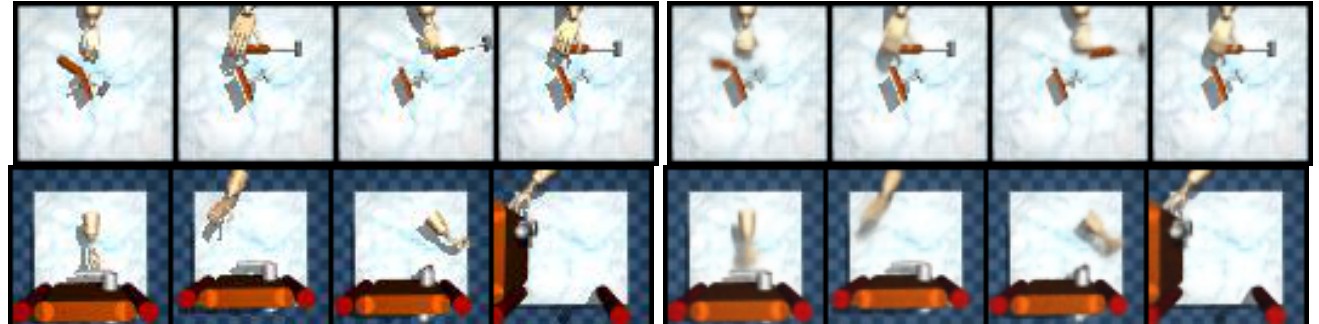

Fig. 11. ROW1: Original input images of the Hammer task; ROW2: Corresponding Reconstructed images; ROW3: Original input images of the Door task; ROW4: Corresponding Reconstructed images. These images depict that the latent features sufficiently encodes features required to reconstruct the images.

For training, we collected a dataset of 1 million images of size 64 x 64. Out of the 1 million images collected, 25% of the images are collected using an optimal course of actions (expert policy), 25% with a little noise (expert policy + small noise), 25% with even higher level of noise (expert policy + large noise) and remaining portion by randomly sampling

[2]https://github.com/aravindr93/mjrl
[3]https://github.com/openai/vdvae

| Parameter | Setting |
|---|---|
| frame_stack | 3 |
| replay_buffer_capacity | 100000 |
| init_steps | 1000 |
| batch_size | 128 |
| hidden_dim | 1024 |
| critic_lr | 1e-3 |
| critic_beta | 0.9 |
| critic_tau | 0.01 |
| critic_target_update_freq | 2 |
| actor_lr | 1e-3 |
| actor_beta | 0.9 |
| actor_log_std_min | -10 |
| actor_log_std_max | 2 |
| actor_update_freq | 2 |
| discount | 0.99 |
| init_temperature | 0.1 |
| alpha_lr | 1e-4 |
| alpha_beta | 0.5 |

TABLE III

SAC HYPERPARAMETERS.

| Environment | action_repeat |
|---|---|
| Cartpole, Swing | 8 |
| Reacher, Easy | 4 |
| Cheetah, Run | 4 |
| Cup, Catch | 4 |
| Walker, Walk | 2 |
| Finger, Spin | 2 |

TABLE IV

ACTION REPEAT VALUES FOR DMCONTROL SUITE

actions (random actions). This is to ensure that the images collected sufficiently represents the distribution faced by policy during the training of the agent. We observed that this significantly helps compared to collecting data only from the expert policy. The variational auto-encoder(VAE) is trained using a reconstruction objective [7] for 10epochs. Figure 11 showcases the reconstructed images. We used a latent size of 512 for a fair comparison with Resnet. The weights of the encoder are freezed and used as feature extractors in place of Resnet in RRL. RRL(VAE) also uses the inputs from the pro-prioceptive sensors along with the encoded features. VAE implementation courtesy [53].

*H. Visual Distractor Evaluation details*

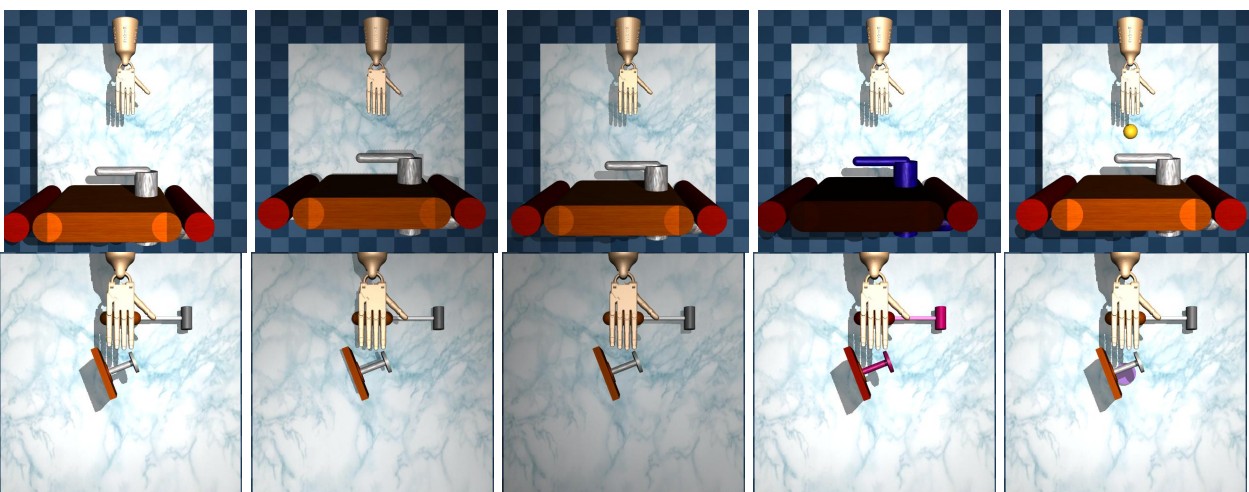

Fig. 12. COL1: Original images; COL2: Change in light position; COL3: Change in light direction; COL4: Randomizing object colors; COL5: Introducing a random object in the scene. All the parameters are randomly sampled every time in an episode.

In order to test the generalisation performance of RRL and FERM [58], we subject the environment to various kinds of visual distractions during inference (Figure 12). Note all parameters are freezed during this evaluation, an average performance over 75 rollouts is reported. Following distractors were used during inference to test robustness of the final policy -

- Random change in light position.
- Random change in light direction.
- Random object color. (Handle, door color for Door-v0; Different hammer parts and nail for Hammer-v0)
- Introducing a new object in scene - random color, position, size and geometry (Sphere, Capsule, Ellipsoid, Cylinder, Box).

*I. Compute Cost calculation*

We calculate the actual compute cost involved for all the methods (RRL(Ours), FERM, RRL(Resnet-50), RRL(Resnet-18)) that we have considered. Since in a real-world scenario there is no simulation of the environment we do not include the cost of simulation into the calculation. For fair comparison we show the compute cost with same sample complexity (4 million steps) for all the methods. FERM is quite compute intensive (almost 5x RRL(Ours)) because (a) Data augmentation is applied at every step (b) The parameters of Actor and Critic are updated once/twice at every step (Compute results shown are with one update per step) whereas most of the computation of RRL goes in the encoding of features using Resnet. The cost of VAE pretraining in included in the over all cost. RRL(Ours) that uses Resnet-34 strikes a balance between the computational cost and performance. **Note:** No parallel processing is used while calculating the cost.