# OpenReview forum: "RRL: Resnet as representation for Reinforcement Learning"
_ICRA.org/2022/Workshop/Contact-Rich — ICRA 2022 Workshop: RL for Manipulation Oral_

### Official Review · Reviewer_qGXA · 2022-05-05

**Rating:** 9
**Confidence:** 4

**Review:**

### Summary
This paper proposed the use of ResNet as a representation for reinforcement learning. The main idea is to use a ResNet34 model pre-trained on ImageNet to provide a sufficiently wide distribution of real-world scenarios. The pre-trained is then frozen and combined with a reinforcement learning & imitation learning approach.

The topic of this paper is very interesting and relevant to the workshop.

### Comments
Although the idea of using a pre-trained model to extract visual features for a RL agent is not new, the authors do a good job at referencing related work and then presenting the case for their proposed method.
- The paper is well organized and clearly explained.
- The experimental evaluations include a variety of baselines and tasks as well as evaluation of different aspects of the proposed method.
- A proper discussion of the strengths and limitations of the proposed method is also included.

### Suggestions:
- Consider more clearly describing the difference between your proposed approach and previous methods that have already explored similar ideas. For example, though [1] is cited as "learning without explicit representation", the idea of using a pre-trained (with ImageNet) visual model was already proposed. Many papers have since then followed ideas similar to [1].
A while ago, I read another similar research [2], where ResNet34 is similarly used to extract visual features of the environment and then use that in a RL framework. [2] then focuses on learning a dense reward to guide the RL agent.

[1] Levine, Sergey, et al. "End-to-end training of deep visuomotor policies." The Journal of Machine Learning Research 17.1 (2016): 1334-1373.
[2] Luo, Yongle, et al. "Balance between efficient and effective learning: Dense2sparse reward shaping for robot manipulation with environment uncertainty." arXiv preprint arXiv:2003.02740 (2020).

---

### Official Review · Reviewer_z6bU · 2022-05-09
**Review of Paper "RRL: Resnet as representation for Reinforcement Learning"**

**Rating:** 8
**Confidence:** 4

**Review:**

**Summary**: This paper proposes a method called RRL, which leverages pre-trained visual representation for Reinforcement Learning. RRL first trains an encoder (Resnet) on the ImageNet dataset using a classification objective. The final layer is then removed and the intermediate features are used as RL policy input. The method is validated on 4 dexterous manipulation tasks and compared with 3 baselines.

**Strengths**
- The method is simple and works very well.
- Experiments are extensive. The results are very good and will serve as a strong baseline for future work.

**Comments**
- It's stated that DAPG (state) performance is the upper bound for RRL, but in Fig. 4, the performance of RRL is better than DAPG(state) in 2 tasks. RRL even learns faster in 3 tasks which is quite unintuitive as RRL has
- In V-H, the authors discussed that representations learned by RAD are task-specific. The advantage of RAD is that it learns representation tailored to a specific task. With this in mind, I think an interesting direction is to extend this method so that the representation can be adapted or fine-tuned online. The pre-trained representation could serve as a good initialization. This would also solve the data mismatch problem.